# Capture-based enrichment of *Theileria parva* DNA enables full genome assembly of first buffalo-derived strain and reveals exceptional intra-specific genetic diversity

Nicholas C. Palmateer[1], Kyle Tretina[1], Joshua Orvis[1], Olukemi O. Ifeonu[1], Jonathan Crabtree[1], Elliott Drabék[1], Roger Pelle[2], Elias Awino[3], Hanzel T. Gotia[1], James B. Munro[1], Luke Tallon[1], W. Ivan Morrison[4], Claudia A. Daubenberger[5,6], Vish Nene[3], Donald P. Knowles[7], Richard P. Bishop[7], Joana C. Silva[1,8]*

**1** Institute for Genome Sciences, University of Maryland School of Medicine, Baltimore, Maryland, United States of America, **2** Biosciences eastern and central Africa-International Livestock Research Institute, Nairobi, Kenya, **3** International Livestock Research Institute, Nairobi, Kenya, **4** The Roslin Institute, Royal (Dick) School of Veterinary Studies, University of Edinburgh, Easter Bush Campus, Roslin, United Kingdom, **5** Swiss Tropical and Public Health Institute, Basel, Switzerland, **6** University of Basel, Basel, Switzerland, **7** Department of Veterinary Microbiology and Pathology, Washington State University, Pullman, Washington, United States of America, **8** Department of Microbiology and Immunology, University of Maryland School of Medicine, Baltimore, Maryland, United States of America

* jcsilva@som.umaryland.edu

**Data Availability Statement:** Raw sequence reads and corresponding genome assemblies are available from NCBI, under BioProject accession

## Abstract

*Theileria parva* is an economically important, intracellular, tick-transmitted parasite of cattle. A live vaccine against the parasite is effective against challenge from cattle-transmissible *T. parva* but not against genotypes originating from the African Cape buffalo, a major wildlife reservoir, prompting the need to characterize genome-wide variation within and between cattle- and buffalo-associated *T. parva* populations. Here, we describe a capture-based target enrichment approach that enables, for the first time, *de novo* assembly of nearly complete *T. parva* genomes derived from infected host cell lines. This approach has exceptionally high specificity and sensitivity and is successful for both cattle- and buffalo-derived *T. parva* parasites. *De novo* genome assemblies generated for cattle genotypes differ from the reference by ~54K single nucleotide polymorphisms (SNPs) throughout the 8.31 Mb genome, an average of 6.5 SNPs/kb. We report the first buffalo-derived *T. parva* genome, which is ~20 kb larger than the genome from the reference, cattle-derived, Muguga strain, and contains 25 new potential genes. The average non-synonymous nucleotide diversity ($\pi_N$) per gene, between buffalo-derived *T. parva* and the Muguga strain, was 1.3%. This remarkably high level of genetic divergence is supported by an average Wright's fixation index ($F_{ST}$), genome-wide, of 0.44, reflecting a degree of genetic differentiation between cattle- and buffalo-derived *T. parva* parasites more commonly seen between, rather than within, species. These findings present clear implications for vaccine development, further demonstrated by the ability to assemble nearly all known antigens in the buffalo-derived strain, which will be critical in design of next generation vaccines. The DNA capture approach used provides a clear advantage in specificity over alternative *T. parva* DNA

numbers PRJNA656576 (T. parva Muguga_BV115), PRJNA656578 (T. parva Uganda), PRJNA656581 (T. parva Marikebuni) and PRJNA656583 (T. parva lawrencei, aka, Buffalo_3081).

**Funding:** Funding for the work was provided by: Bill and Melinda Gates Foundation (US) (OPP1078791); Agricultural Research Service (59–5348–4-001); National Institute of Allergy and Infectious Diseases (R01AI141900). The funders had no role in study design, data collection and analysis, decision to publish, or preparation of the manuscript.

**Competing interests:** The authors have declared that no competing interests exist.

enrichment methods used previously, such as those that utilize schizont purification, is less labor intensive, and enables in-depth comparative genomics in this apicomplexan parasite.

## Author summary

An estimated 50 million cattle in sub-Saharan Africa are at risk of the deadly livestock disease East coast fever (ECF), caused by the parasite *Theileria parva*, which imposes tremendous economic hardship on smallholder farmers. An existing ECF vaccine protects against strains circulating among cattle, but not against *T. parva* derived from African Cape buffalo, its main wildlife carrier. Understanding this difference in protective efficacy requires characterization of the genetic diversity in *T. parva* strains associated with each mammalian host, a goal that has been hindered by the proliferation of *T. parva* in nucleated host cells, with much larger genomes. Here we adapted a sequence capture approach to target the whole parasite genome, enabling enrichment of parasite DNA over that of the host. Choices in protocol development resulted in nearly 100% parasite genome specificity and sensitivity, making this approach the most successful yet to generate *T. parva* genome sequence data in a high-throughput manner. The analyses uncovered a degree of genetic differentiation between cattle- and buffalo-derived genotypes that is akin to levels more commonly seen between species. This approach, which will enable an in-depth *T. parva* population genomics study from cattle and buffalo in the endemic regions, can easily be adapted to other intracellular pathogens.

## Introduction

In developing countries, infectious diseases of livestock can have a broad and profound negative effect on public health, including malnutrition, increased susceptibility to disease, female illiteracy and loss of productivity [1–3], and curb the potential for economic improvement [4]. *Theileria parva* is a tick-transmitted, obligate intracellular apicomplexan parasite that causes East Coast fever (ECF), an acute fatal disease of cattle in eastern, central and southern Africa. During proliferation of *T. parva* in the mammalian host, a multi-nucleated schizont immortalizes infected host lymphocytes and divides in synchrony with them, ensuring that the infection is transmitted to each daughter cell, through poorly understood mechanisms [5–9]. Susceptible animals usually die within three to four weeks post-infection. This is a result of widespread lysis of infected and uninfected lymphocytes in the lymphoid tissues, secondarily inducing a severe macrophage response, characterized by high IL-17 expression, and pulmonary edema [10]. ECF represents a severe economic constraint and is a major impediment to the development of the cattle industry in the impacted region. The most recent estimate, dating from the 1990's, placed the losses from ECF at a million cattle each year; currently, an estimated 50 million cattle are at risk and annual losses are estimated in $596 million [11–14].

An infection-and-treatment method (a.k.a ITM) involving administration of a lethal dose of a cryopreserved stabilate of *T. parva* sporozoites from three parasite isolates, together with a long-acting formulation of oxytetracycline, has been in use for several decades. Incorporation of the three parasite isolates (known as the Muguga cocktail) is required to avoid vaccine evasion, a common problem of anti-parasitic vaccines [15–17]. This vaccination method can protect against ECF for at least 43 months, although, in a variable percentage of animals, heterologous parasites may induce transient clinical symptoms in vaccinated cattle [18].

However, ITM also has significant drawbacks, including a logistically intensive manufacturing process [reviewed in 19]. Also, there have been recently verified concerns that ITM is not always as effective against challenge from buffalo-derived *T. parva* as it is against cattle-derived parasites [20], and even cattle-derived *T. parva* from geographically diverse regions could sometimes break through immunity induced by the Muguga cocktail vaccine [21]. The African buffalo (*Syncerus caffer*) is an asymptomatic wildlife carrier of *T. parva* in the region and is the primary mammalian host [22]. Areas where buffalo and cattle co-graze enable transmission of the parasite between mammalian hosts by the tick vector, *Rhipicephalus appendiculatus* [23].

Studies using a limited set of markers strongly suggest that the *T. parva* strains (or geno-types) circulating in the affected cattle population represent only a subset of a much more het-erogeneous *T. parva* meta-population residing in buffalo [24–27], due primarily to lack of tick transmissibility of buffalo-derived infections, associated with very low piroplasm counts. *T. parva* isolates obtained from buffalo were at one time classified into a separate subspecies, *T. parva lawrencei*, based on clinical presentation, despite the lack of genetic evidence to support this claim [13]. Preliminary data suggests that genome-wide differences between the reference Muguga strain and buffalo-derived isolates are substantially larger than among cattle-trans-missible genotypes [28], although there is currently no genome assembly for *T. parva* from buffalo. The design of a vaccine that is effective against most cattle- and buffalo-derived *T. parva* requires the comprehensive characterization of genetic differences within and between those two *T. parva* parasite populations, particularly in regions of the genome that encode anti-genic proteins. Comprehensive knowledge of genetic variation in the species is also needed to monitor the impact of live vaccination on the composition of parasite field populations.

The biology of *T. parva* has so far proved a powerful obstacle to the acquisition of DNA in sufficient quantity and quality for whole genome sequencing. DNA extracted from cattle blood early in the infection cycle is heavily contaminated with host DNA. In late stages of infection, the tick-infective piroplasm stage infects erythrocytes, and requires collection of large volumes of blood from clinically ill *T. parva*-infected animals to obtain purified piro-plasm DNA in sufficient quantity for genome sequencing [29], an approach not sustainable or ethically feasible for a large number of strains. In addition, and despite their higher virulence to cattle, *T. parva* of buffalo origin induce lower levels of schizont parasitosis, and produce no or very few piroplasms in cattle [30], precluding their use as a source of parasite DNA. *T. parva* DNA can also be obtained from schizonts purified following lysis of infected lympho-blasts [31, 32] but low yield, host DNA contamination, and the heterogeneity in lysing proper-ties of infected cells make this approach unsuitable for high-throughput applications.

Finally, the estimation of genome-wide population genetic diversity relies on the identifica-tion of sequence variants from the alignment of whole genome sequence data to a reference genome [33]. The same approach has been used to identify pathogen-encoded antigens, which are potential vaccine candidates, since they are often among the most variable protein-coding genes in a genome [34]. However, in highly polymorphic species such as *T. parva*, this approach is unreliable because sequence reads fail to map between strains, particularly in the genomic regions that encode the most variable antigens [28, 35].

Here, we applied a target DNA sequence capture approach to selectively enrich parasite DNA in samples obtained from *T. parva*-infected bovine lymphocyte cultures, consisting mostly of bovine DNA [36]. Even though conceptually similar to pathogen DNA enrichment approaches used before for other organisms [37–40], design choices in the current study resulted in extremely high capture sensitivity and specificity. Furthermore, to gain access to variable genomic regions that cannot be analyzed through read mapping approaches [35], we assembled the captured sequence read data and analyzed the resulting *de novo* genome assem-blies for completeness. Starting from cell cultures in which the parasite DNA was less than 4%

[36], we have generated *de novo* genome assemblies for each isolate consisting of 109–126 scaffolds, that encompass >95% of the reference genome of *T. parva*. This approach was successful even when applied to a highly divergent *T. parva* isolate from buffalo, for which we present the first publicly available genome assembly. The ability to characterize genome-wide polymorphism based on whole genome assemblies, which provide higher resolution relative to read mapping approaches, particularly in highly variable regions of the genome, represents a powerful approach for the characterization of genetic variation in intracellular parasites such as *Theileria*, and in particular for the study of highly polymorphic antigens and other variable genes and regions of the genome.

## Methods

### Ethics statement

The Institutional Animal Care and Use Committee (IACUC) of the International Livestock Research Institute (ILRI) was established in 1993 to ensure that international standards for animal care and use are followed in all ILRI research involving use of animals. The original studies in which cattle were infected, over two decades ago, were specifically approved by ILRI's IACUC. The expansion of the infected lymphocyte cultures, conducted to generate the material used in this study, does not necessitate explicit IACUC approval.

### Samples and parasite-host ratio

Four *T. p*arva isolates were used; they were described originally in Morzaria *et al.* (1995), and have been maintained in culture at the International Livestock Research Institute for over two decades. These schizont-infected bovine lymphocyte cultures were derived from lymph node biopsies taken from cattle experimentally infected with *T. parva* sporozoite stabilates BV115, Marikebuni_3292, Uganda_3645 and Buffalo_7014_3081 (S1 Fig). Stabilate BV115 was established in 2000, the result of a series of stabilates originally derived from the Muguga isolate, obtained circa 1960 when field ticks were fed on cattle at the National Veterinary Research Center (NVRC; now the Veterinary Research Institute), in Muguga, Kenya. BV115 is a cloned stabilate. The Uganda and Marikebuni stabilates were also clonal, either cloned themselves or derived from cloned stabilates (Morzaria et al. 1995), whereas the Buffalo_7014 stabilate was not cloned and therefore could contain multiple parasite genotypes (S1 Fig). Bovine lymphocytes infected with the schizont stage of each isolate were propagated using established protocols [41]. DNA was extracted from schizont-infected lymphocyte cell line cultures using standard protocols [42], including lysis with SDS and proteinase K digestion, followed by phenol/chloroform extraction and ethanol precipitation. The ratio of parasite to host DNA was estimated for each sample, using a qPCR-based approach to estimate the absolute DNA amount separately for bovine and *T. parva* DNA [36].

### Genomic library construction

Library preparation was initiated using 900–1200 ng of total DNA, generated from the extraction of total DNA from infected lymphocyte cultures (S2 Table). Paired-end (PE) genomic DNA libraries were constructed for sequencing on Illumina platforms using the NEBNext DNA Sample Prep Master Mix Set 1 (New England Biolabs, Ipswich, MA). First, DNA was sheared with the Covaris E210, to fragments targeted to 500–700 bp in length. Then libraries were prepared using a modified version of manufacturer's protocol. The DNA was purified between enzymatic reactions and the size selection of the library was performed with AMPure XT beads (Beckman Coulter Genomics, Danvers, MA).

## Whole-genome DNA sequence capture

A custom-designed Nimblegen SeqCap EZ oligo library was used to target capture the *T. parva* genomic DNA in each genomic library for high-throughput sequencing using the 454 GS FLX and Illumina HiSeq2000 and MiSeq platforms. The capture method utilizes custom-designed, biotinylated oligonucleotides for hybridization to the target sequence. The custom oligo library used here was designed based on the *T. parva* Muguga reference genome sequence with accession number AAGK01000000. The probes are proprietary which did not allow quantification of capture efficiency given the number of mismatches to the probe. However, according to Roche, while the efficiency of pulldown depends on sequence context, secondary structure, %GC, and the position of the mismatches/indels within the sequence, the method is robust to at least 10% sequence divergence between probe and target sequences and as little as a 30 consecutive nucleotide match may pull down a fragment. Following hybridization of library fragments to the oligo baits, streptavidin-coated magnetic beads are used to capture the bound fragments, and unbound fragments are washed away leaving captured library fragments ready for sequencing.

## Sequencing

The BV115 Illumina PE library was sequenced using the 100 bp paired-end protocol on an Illumina HiSeq2000 sequencer, using approximately 7.25% of a flowcell lane. The libraries for the remaining three isolates were sequenced on a MiSeq platform, using the 250 bp paired-end protocol, multiplexed into a single run, with each using roughly 1/3 of the sequencing capacity. Raw data from the sequencers was processed using Illumina's RTA and CASAVA pipeline software, which includes image analysis, base calling, sequence quality scoring, and index demultiplexing. Data was then processed through both FastQC (http://www.bioinformatics. bbsrc.ac.uk/projects/fastqc/) and in-house pipelines for sequence assessment and quality control. These pipelines report numerous quality metrics and perform a megablast-based contamination screen. By default, our quality control pipeline assesses base call quality and truncates reads where the median Phred-like quality score falls below Q20.

## Read mapping and genome assembly

The Illumina sequence data were aligned to the reference genome (accession number AAGK01000000) using the Bowtie 2 aligner [43]. Statistics for depth of coverage (number of reads mapped per position) and reference genome breadth of coverage (fraction of the reference genome to which reads map) were generated using internal protocols. Gene coverage was calculated using genomecov from the bedtools suite of tools [44]. The Illumina data were assembled using the SPAdes Assembler v3.9.0 [45]. Because of coverage limitations inherent to the assembler software, and to avoid overrepresentation of sequencing errors and introduction of erroneous duplications, the high-coverage Illumina data was randomly sub-sampled to depths of coverage ranging between 10X and 200X of the reference genome assembly in 10X and 25X increments. The optimal assembly in each case was selected using a combination of statistics including total contig count, contig N50 (contig length for which the set of all contigs of that length or longer contains at least half of the assembly), maximum contig length and total assembly length. The optimal assembly for the BV115 and Uganda assemblies had read coverage cutoff values of 25X, and the optimal Marikebuni and Buffalo_3081 assemblies had read coverage cutoff values of 10X. Assembly contigs were evaluated for host contamination and any contigs matching to the host were removed. In-house scripts were used to determine the extent of overlap between new assemblies and the reference genome (reference genome breadth of coverage) and to generate statistics regarding genes present or absent from the new

assemblies. Assembly correction was done using Pilon (v1.22) [46], using default parameters, with the respective Illumina sequencing reads for each strain. Assembly gene coverage for each strain was calculated using in-house scripts (https://github.com/jorvis/biocode/blob/master/general/calculate_gene_coverage_from_assembly.py).

## Single nucleotide polymorphism (SNP) and structural variant detection and characterization

The Bowtie 2 alignments were converted to a BAM file using SAMtools [47]. The Genome Analysis Toolkit (GATK) [48] was used to identify and correct misalignments caused by small indels, and then to call both SNPs and indels. The resulting VCF file was used to call the major allele (https://github.com/igs-jcsilva-lab/variant-calling-pipelines/blob/master/scripts_for_driver/calling_majorallele.pl), and was then filtered with stringent criteria to eliminate potentially false SNPs, requiring depth greater than 12, quality greater than 50, phred-scaled p-value using Fisher's Exact Test less than 14.5, and Root Mean Square mapping quality zero less than 2. The SNPs were classified by location into intergenic, intronic, synonymous, non-synonymous, read-through or non-sense using VCFannotator (vcfannotator.sourceforge.net). SNPs were detected in assemblies using the show-snps option of the MUMmer3 (v3.23) [49]. The updated *T. parva* Muguga genome annotation [50] was transferred on to the new assemblies using the Genomic Mapping and Alignment Program (GMAP) v2014-04-06 [51]. Assembly-tics was used to identify structural variants between the *de novo* assemblies and the reference genome or corresponding assemblies generated from 454 sequencing [52].

## Nucleotide diversity estimation and genetic differentiation between populations

Nucleotide diversity (the average number of nucleotide differences per site, $\pi$), was calculated based on SNPs called from read mapping and from assembly comparison. Nucleotide diversity from read mapping was done using the VCFtools v0.1.14 package with the—site-pi option [53]. This approach requires reads to map across isolates and does not correct for multiple hits. Nucleotide diversity from *de novo* assemblies was estimated for the CDS alignment for each gene, using the Nei-Gojobori method, which corrects for multiple hits [54]. In our study, this approach requires that the locus be present in the *de novo* assemblies.

Genetic differentiation between cattle- and buffalo-derived *T. parva* populations was estimated with Wright's fixation index, $F_{ST}$ [55], as implemented by Weir and Cockerham [56]), using VCFtools v0.1.14 [53]. $F_{ST}$ measures the proportion of genetic variation explained by population differentiation, and varies between 0, for panmictic population, to 1, in fully differentiated populations.

## Gene prediction

To identify novel genes in contigs and genomic segments with no gene annotations we used the gene prediction software Genemark-ES [57], which according to our experience is the most accurate *ab initio* gene prediction software for *T. parva* [50]. To ensure that these contigs are truly part of the *T. parva* genome, we considered only predicted genes that were contained within contigs encoding homologs to *T. parva* genes. To identify true genes, we used BLASTN to search each predicted gene model against NCBI's non-redundant nucleotide database [58], selecting only those with E-value less than $1\times10^{-5}$. Genes matching to multigene family members were removed from consideration as novel genes. Prediction of transmembrane helices in proteins was done using TMHMM2.0 [59]. Prediction of GPI-anchor sites was done using

PredGPI [60]. The presence and location of signal peptide cleavage sites in amino acid sequences was predicted using SignalP 4.1 [61].

## Results

### Design of the whole-genome sequence capture approach: Length and genome coverage of the probe set, and genomic library fragment size

We have customized a DNA sequence capture approach to obtain *T. parva* genomic DNA from *T. parva*-infected bovine lymphocyte cell lines. The premise is similar to that of exome capture [62, 63] in that the target DNA is only a small subset of the total DNA mix, but here the DNA fraction intended for capture is the 8.31 Mb-long *T. parva* nuclear genome and the 39 kb-long apicoplast genome, while the non-target DNA is the >300 times larger animal host genome. The capture probe set almost completely spans the nuclear and apicoplast genomes of the reference *T. parva* Muguga [29], and are based on the SeqCap EZ platform (Roche/Nimblegen). The probe design was conducted by Roche/Nimblegen using proprietary software. As part of the probe design, probe length was minimized, to increase the success of cross-strain DNA capture of loci with highly diverged segments by taking advantage of relatively small, conserved DNA segments that are intermixed with more rapidly evolving regions; the resulting probes average 76 bp in length. Probes that mapped to low complexity sequences or >5 genomic regions were eliminated, as were those with strong sequence similarity to the bovine genome. The final probe set consists of a series of overlapping probes, which cover 7,932,549 bp (95.5%) of the combined length of the nuclear genome (8,308,027 bp) and the apicoplast genome (39,579 bp). The fraction of the two genomes not covered by probes is spread among 3,843 independent genomic regions that average 93 bp in length (Fig 1, S1 Table). In total, 53 genes have no probe coverage, while 4,111 genes have at least some coverage by the probe set, with >50% of all genes being completely covered by probes (S1 Table). The 53 genes without probe coverage are all members of multigene families, including *Theileria parva* repeat (*Tpr*) and Subtelomere-encoded Variable Secreted Protein (SVSP). To maximize the probability of capturing both genes that are highly variable across strains and genomic regions without probe coverage, we sheared the genomic DNA sample to a fragment size significantly larger

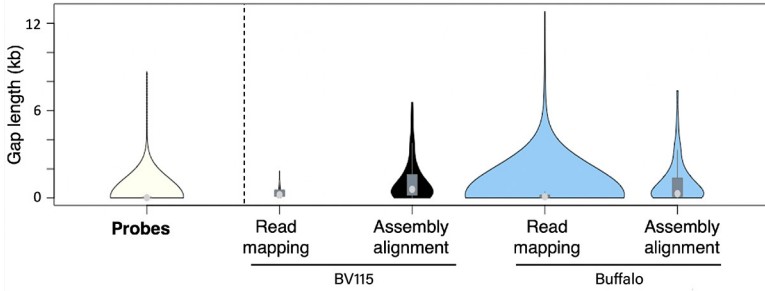

**Fig 1. Gaps in coverage of reference genome for capture probes, reads and assemblies.** Violin plots of length distribution of gaps in coverage of *T. parva* reference genome by capture probes (n = 3,843), as well as for sequence data generated for BV115 (Muguga) and Buffalo_3081 (buffalo-derived) *T. parva* isolates. For each isolate, gaps in coverage were identified after mapping of sequence reads (Read mapping) and after alignment of *de novo* assemblies (Assembly alignment) to the reference *T. parva* genome. For BV115, gaps in coverage after read mapping (n = 42) were fewer than observed for Buffalo_3081 (n = 735). The number of gaps after alignment of assembly contigs to the reference genome were similar in BV115 (n = 73) and Buffalo_3081 (n = 68). The Buffalo_3081 sample represents the isolate least similar to the reference and, therefore, to the probes. For the Buffalo_3081 isolate, and in contrast with BV115, assembly alignment resulted in fewer and smaller gaps than read mapping. The median is shown by the light gray circle, and the interquartile range is shown with the dark gray rectangle.

than the length of the probes, such that captured DNA fragments contain both conserved segments that hybridize to the probes and highly variable regions that flank them.

## Data generation

Four *T. parva* isolates were used in this study, which were described in earlier studies [36, 64]. Briefly, cell line BV115 consists of infected lymphocytes resulting from the experimental infection of *Bos taurus* animal BV115 with a clone of the original Muguga reference stock (S1 Fig). This parasite, obtained from the Kenyan highlands, is the source of the reference *T. parva* genome [29], and was also the template for the design of the capture probes. Therefore, enrichment results using the BV115 isolate represent both a proof of principle for this approach and positive control, representing the best possible scenario of a perfect sequence match between probes and the DNA they target. Two other clones derived from cattle infections were used, namely *T. parva* Marikebuni (stabilate 3292), from coastal Kenya, and Uganda (stabilate 3645), from northwest Uganda. These three isolates, originally obtained from cattle, are henceforth designated as "cattle-transmissible" or "cattle-derived". To determine the success of this approach for buffalo-derived *T. parva*, we used an isolate from Buffalo 7014 (stabilate 3081), originally derived from an African Cape buffalo, which we refer to as Buffalo_3081. Library preparation was initiated using 900–1200 ng of total DNA, generated from the extraction of DNA from infected lymphocyte cultures (S2 Table). The proportion of *T. parva* DNA in each sample was 1.9%, 3.1%, 0.9% and 1.7% for BV115, Marikebuni, Uganda and Buffalo_3081, respectively, with the remainder being host DNA [36] (**Table 1**).

For each sample, the gDNA shearing length targeted was 500–700 bp, with average size of captured fragments between 446 and 619 bp. Libraries were sequenced with either Illumina HiSeq 2000 or MiSeq platforms, and 5,687,838 to 12,174,316 sequence reads were generated for each sample and mapped to the *T. parva* reference genome (S2 Table).

## Approach specificity, sensitivity and accuracy

The specificity of the capture approach used is defined here as the fraction of the BV115 sequence reads that map to the parasite reference genome. Specificity of the approach was very high, with >98% of BV115 reads generated mapping uniquely to the *T. parva* reference genome (Table 1), a nearly-perfect, 50.5-fold enrichment for parasite DNA. The remaining 1.97% of mapped reads in the present study originated from the host genome.

Sensitivity of the approach is defined here as the fraction of the reference genome to which the BV115 sequence reads map. The sensitivity of this capture approach, based on this probe

**Table 1. Specificity and sensitivity of the capture-based parasite DNA enrichment approach.**

| Isolate | Pre-enrichment *Theileria* gDNA (%)[a] | Total reads generated[b] | Total reads aligned | Mean coverage[c] | Specificity (%)[d] | Sensitivity (%)[e] |
|---|---|---|---|---|---|---|
| BV115 | 1.94 | 12,174,316 | 11,952,650 | 146X | 98.03 | 99.80 |
| Marikebuni 3292 | 3.05 | 7,204,556 | 6,740,297 | 202X | 96.47 | 97.70 |
| Uganda 3645 | 0.92 | 5,687,838 | 5,313,295 | 157X | 97.52 | 98.34 |
| Buffalo_3081 | 1.72 | 6,080,972 | 5,446,541 | 160X | 96.40 | 97.59 |

[a] Proportion of the original DNA sample that is composed of *T. parva* DNA, as measured by qPCR [36].

[b] Read length for BV115 was 101 bp and 250 bp for the other three strains.

[c] Estimation: (Total_reads_aligned*Mean_read_length)/genome_size. The genome size used was the sum of the nuclear and apicoplast reference genomes targeted.

[d] Percent reads generated that mapped to *T. parva* reference genome.

[e] Percent of the *T. parva* reference genome to which reads map.

set, is 99.8%. Not only were all regions of the reference genome against which probes were designed recovered but, in fact, the fraction of the reference genome that is covered by Illumina sequence reads is larger than 95.5%, the fraction of the genome that is covered by probes (Fig 2, Table 1). This result demonstrates that we successfully captured segments of the genome that are not included in the probe set, as intended with the experimental design described above. As a result of the high sensitivity of this approach, despite the 3,843 gaps in probe coverage of the *T. parva* genome, the number of segments of the genome with 0X coverage from read mapping was 42, with average length of 402.6 bp (Fig 1). This was because repeats or low complexity regions eliminated from the probe set were captured in fragments that also contained neighboring unique regions for which capture by the probes was efficient.

The generation of whole genome sequence (WGS) data based on this capture approach includes one amplification step. To verify the accuracy of the WGS data, and in particular to assess its error rate, we mapped the sequence reads against the reference Muguga genome assembly [29], and identified SNPs. Despite the differences in library protocol, sequencing platforms and data preparation we identified only 107 SNPs across the entire 8.31 Mb genome, for a SNP density of $\sim 1 \times 10^{-5}$ SNPs/bp, below the sequencing error of the Illumina platform, and thus providing independent confirmation of the quality of the data generated here (S3 Table).

## Capture specificity and sensitivity for non-reference strains

Both specificity (96.4% - 97.5%) and sensitivity (97.6% - 98.3%) were very high albeit slightly lower for non-reference *T. parva* isolates than they were for BV115 (Table 1). However, these values are likely underestimates. Some *T. parva* genes, including those coding for known antigens and some proteins involved in host-parasite interactions, are known to be highly variable, with polymorphism >2%, and possibly much higher [27]. This level of polymorphism poses potential challenges at two levels: *i*) decreased capture efficiency with increasing sequence divergence between probes and target sequences, and (*ii*) lack of read mapping beyond a sequence divergence threshold between sequence reads and genome reference. The former will result in decreased sensitivity while the latter will result in underestimation of both specificity and sensitivity.

To determine which of these two factors is responsible for the lower sensitivity in the three non-reference isolates relative to BV115, we generated *de novo* genome assemblies for each isolate, based on the sequence capture data, and compared coverage of each Muguga locus by read mapping to the completeness of the locus sequence extracted from each *de novo* assembly (see section on "Sequence variant detection" below). Draft genome assemblies for the non-reference cattle-derived *T. parva* strains used in this study (namely, Marikebuni and Uganda) were generated before [65]. These were generated with DNA obtained from purified piroplasms collected from blood of animals infected with each clonal cell line, which was then sequenced with 454 pyrosequencing technology and assembled. Hence, the input DNA consisted of the complete genome of each strain and these draft assemblies should be fairly complete, except potentially for repetitive regions that could not be completely resolved. These assemblies enabled the estimation of sensitivity of the capture approach relative to each orthologous reference genome, which corresponds to the percent of the respective reference genome with coverage by reads obtained from capture in the current study, which was 98.6% for Marikebuni and 99.3% for Uganda (S4 Table). The vast majority of the missed segments are likely either subtelomeric repeats or members of repeat families that could not be unambiguously assembled.

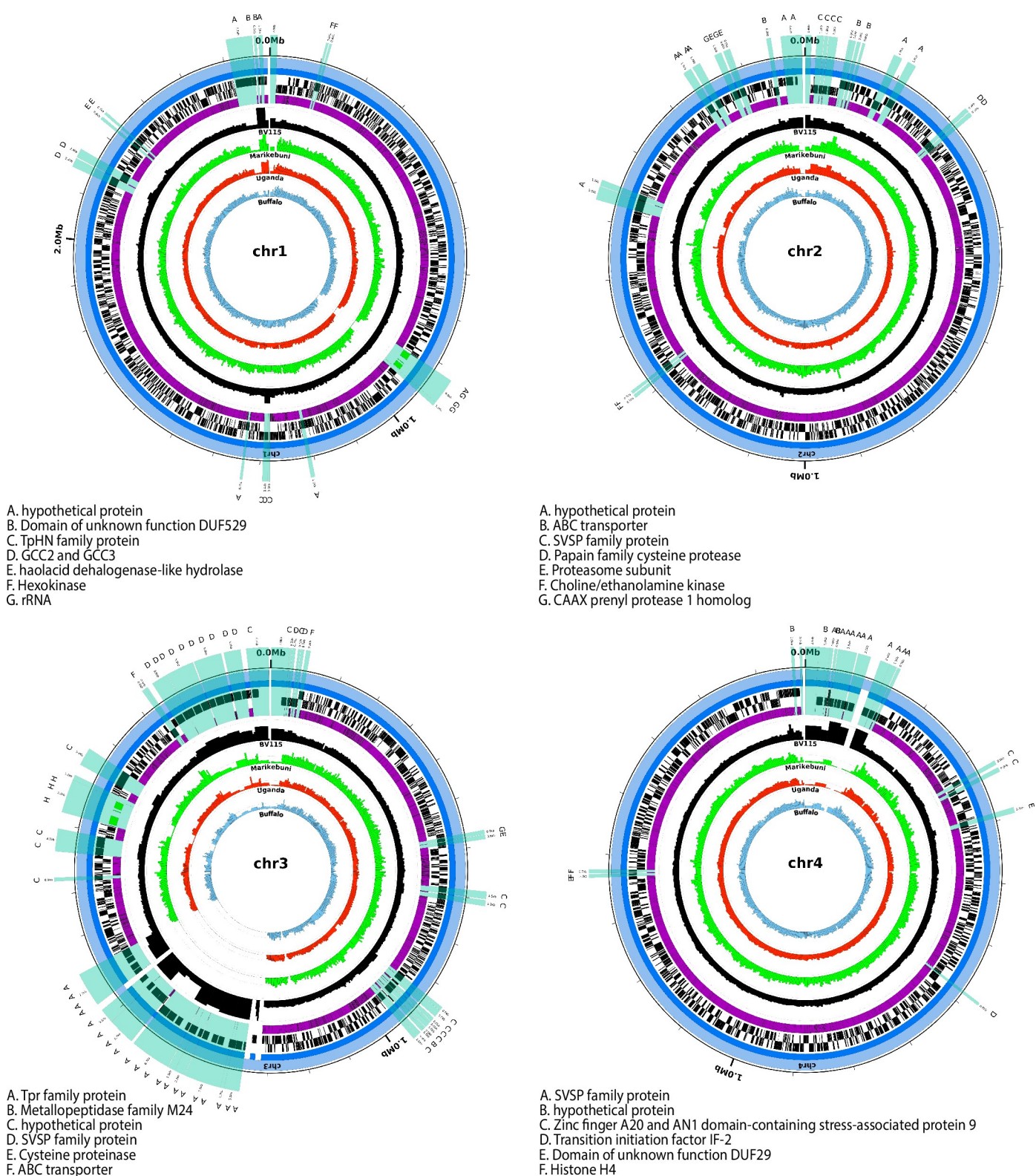

A. hypothetical protein
B. Domain of unknown function DUF529
C. TpHN family protein
D. GCC2 and GCC3
E. haolacid dehalogenase-like hydrolase
F. Hexokinase
G. rRNA

A. hypothetical protein
B. ABC transporter
C. SVSP family protein
D. Papain family cysteine protease
E. Proteasome subunit
F. Choline/ethanolamine kinase
G. CAAX prenyl protease 1 homolog

A. Tpr family protein
B. Metallopeptidase family M24
C. hypothetical protein
D. SVSP family protein
E. Cysteine proteinase
F. ABC transporter
G. Papain family cysteine protease
H. rRNA

A. SVSP family protein
B. hypothetical protein
C. Zinc finger A20 and AN1 domain-containing stress-associated protein 9
D. Transition initiation factor IF-2
E. Domain of unknown function DUF29
F. Histone H4

**Fig 2. Alignment of probe coverage and read mapping in *T. parva* nuclear chromosomes.** For each of the four *T. parva* nuclear chromosomes, starting from the outer-most circle, the following tracks are shown: chromosome scaffolds (light blue); assembly contigs (dark blue); genes encoded in the forward strand, including

protein-coding (black), rDNA (green) and tRNA (red) genes; genes encoded in the reverse strand (protein-coding, rDNA, and tRNAs); regions covered by probes (purple); BV115 coverage (absolute read counts in 5kb windows; maximum shown is 500) (black); Marikebuni coverage (green); Uganda coverage (orange); Buffalo_3081 coverage (sky blue). Chromosomal regions without probes that are ≥500 bp are magnified 5000X and highlighted in transparent light green across the tracks showing chromosome, contig, forward and reverse strand genes, and probe coverage. The proteins encoded within these regions without probes are labeled, with the key for each chromosome listed on the bottom left of each plot.

### *De novo* genome assemblies based on whole genome sequence capture data

We built several assemblies with the BV115 data, varying the assembly software and genome depth of coverage, and selected the assembly with the longest N50 and cumulative length. For the reference strain, represented by BV115, nearly every gene was represented in full in the *de novo* genome assembly, and only 7 genes (<0.2% of all genes) were completely absent (S5 Table). The missing genes are located in the most probe-poor regions of the genome, and mainly consist of SVSP multigene family members [66]. In contrast to the assembly data, when the BV115 sequence reads were mapped directly to the reference genome there were no nuclear genes completely or partially missing, demonstrating that despite gaps in probe coverage, we were able to obtain complete nuclear gene coverage in our positive control isolate, and that the fact that some of these genes are missing in the assembly may be due to the difficulty to unambiguously assemble these regions. The *Tpr* locus, which spans a central region of chromosome 3, is represented in the reference genome assembly by two small contigs plus the edge of one of the two larger chromosome 3 contigs [29]. Interestingly, despite the near complete lack of probes in the *Tpr* locus, in BV115 we were able to capture reads that map to most of the locus (Fig 2) and reconstruct partially assembled contigs for this chromosomal region (Fig 3). This suggests that, in the reference Muguga strain (used to infect animal BV115, and the

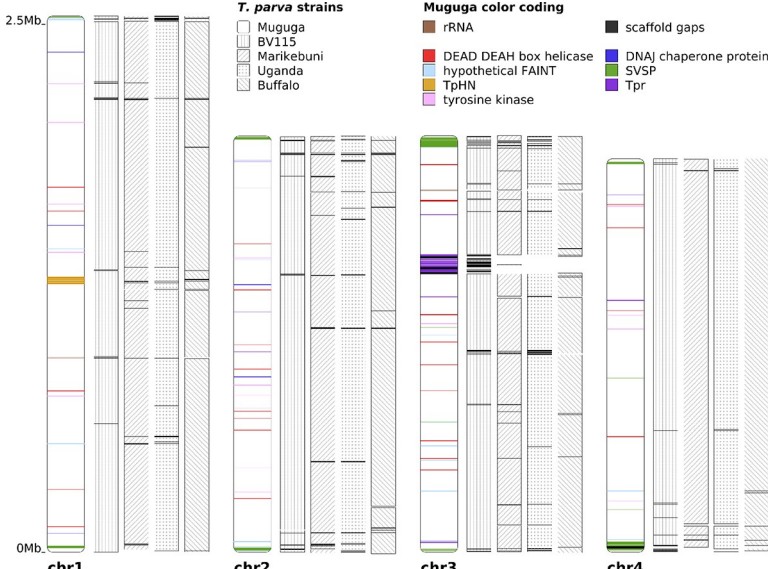

**Fig 3. *De novo* genome assemblies for four *T. parva* isolates.** In the reference assembly (Muguga), chromosomes 1 and 2 are represented by single scaffolds. The four contigs representing chromosome 3 and the two contigs of chromosome 4 were each merged and are shown as a single scaffold. Assemblies for the four new isolates were generated with SPAdes, and aligned to the *T. parva* Muguga reference, using nucmer. Scaffold placement was centered on the midpoint of each alignment. All gene families with >10 paralogs are colored in the reference Muguga strain. The main genomic regions underrepresented in the new assemblies contain several members of *T. parva* multigene families, including the *Tpr* locus (a total of 24 *Tpr* genes within the two contigs central in chromosome 3), and SVSP family (15 SVSP genes in the first 20kb in the 5′ end of chromosome 4, shown at the bottom).

reference used for probe design), either a *Tpr* gene outside of the *Tpr* locus is similar enough in sequence for probes based on its sequence to cross-capture the *Tpr* locus, or that the few probes present in this *Tpr* locus region were sufficient to anchor and capture sequence fragments that span most of this region of chromosome 3.

*De novo* assemblies were also generated for the three non-reference genotypes. They each consist of <130 scaffolds and encompass >95.6% of the reference *T. parva* genome assembly (Fig 3; S5 Table). In each case, fewer than 2% of the nuclear genes were completely missing: 56 in Marikebuni, 31 in Uganda, and 27 in Buffalo_3081. One notable area lacking assembly coverage in all three genome assemblies was the *Tpr* locus. While these genes were not difficult to capture in the BV115, read coverage is poor in the non-reference isolates, suggesting that the reference-based probes failed to capture these genes in other genomes, consistent with the hyper variability of this gene family [67]. Of the 39 *Tpr* genes, including dispersed copies present throughout the genome, only six were recovered either partially or completely in all three non-reference genotypes. For each strain, there were several contigs that were not incorporated into the respective genome assembly. These contigs contain sequences with homology to a number of *T. parva* genes, a majority of which are members of multigene families or hypothetical proteins (S6 Table).

## Sequence and structural accuracy of *de novo* assemblies

To correct possible errors and validate the *de novo* assemblies, a number of steps were taken. Illumina reads for each of the four strains were mapped to their respective *de novo* assembly. The number of nucleotide differences detected varied between 73 in BV115 and 431 in Buffalo_3081. Assemblies were then polished with Pilon [46] which corrected several of the differences identified (S4 Table). The remainder may represent non-specific read mapping in regions of multigene families, regions with insufficient read coverage for sequence correction or variants segregating in culture at the time of DNA isolation.

The draft assemblies for the non-reference cattle-derived *T. parva* strains used in this study, generated before with 454 data [65], allowed validation of the sequence data and assemblies generated here with a capture-based approach. Alignment of the Illumina reads to the 454-based assemblies yielded 203, and 79 SNPs respectively for the Marikebuni and Uganda strains, for a density of $<1–2 \times 10^{-5}$ differences per bp. In addition, the BV115 reads were mapped against the Muguga reference genome, with 107 differences identified. All these values fall within the margin of error of Illumina sequencing, again consistent with the high quality of the data generated. Alignment of our *de novo* assemblies to the previously generated 454-based assemblies resulted in a similarly low number of SNPs in each of the three strains (S4 Table).

Finally, when comparing the structure of the BV115 assembly to the Muguga reference, a total of 26 structural variants were detected, with a cumulative length of 11,388 bp (S7 Table). The Marikebuni and Uganda *de novo* assemblies were compared to their respective reference assemblies generated from 454 sequencing data [65]. The 454-based assemblies consist of 985 and 507 contigs, respectively for Marikebuni and Uganda. These comparisons each yielded 11 or fewer structural variants, totaling at most ~1000 bp in length (S7 Table). We observed a considerably larger number and cumulative length of structural variants in BV115 relative to its reference compared to what is observed for the other two cattle-derived genotypes, which is possibly an artifact of the high fragmentation of the Marikebuni and Uganda 454 sequencing-based reference genomes (which may prevent the detection of structural differences) relative to the nearly closed Muguga reference. The very low number of cumulative base pairs affected by these variants highlight the high accuracy of the *de novo* assemblies build with the capture data.

## First genome assembly for a buffalo-derived *T. parva* parasite

The generation of *de novo* genome assemblies allows a comprehensive characterization of differences in homologous genomic regions between new strains and the reference. It also makes it possible to characterize missing and unique regions in the new genomes, provided that those regions flank regions that match probes, or that they represent duplicated regions with high sequence similarity to probe-covered genomic segments in the reference. In these situations, *de novo* assemblies also provide a clear advantage for genome characterization over read mapping approaches.

The alignment of the Buffalo_3081 assembly to the Muguga reference revealed >300 structural variants in regions of sequence homology between genomes, affecting a cumulative length of 128,750 bp. These included insertions/expansions totaling a gain of 61,632 bp (83 insertions totaling 17,534 bp in length, 26 tandem duplications totaling 20,000 bp, and 71 repeat expansions totaling 24,098 bp) as well as structural changes resulting in genome reduction, namely deletions and contractions amounting to a loss of 67,118 bp (S8 Table). This suggests that rates of repeat expansion and contraction are fairly balanced.

Overall, however, the genome assembly of the buffalo-derived isolate (Buffalo_3081) is 8,366,826 bp long (S5 Table), ~20 Kb longer than the reference *T. parva* Muguga assembly, despite missing genomic regions for which probes were not designed. Compared to that of BV115, which was used as a positive control, the Buffalo_3081 assembly is approximately 130 Kb longer, which supports the hypothesis that the Buffalo_3081 genome is indeed longer. Therefore, we sought to characterize regions unique to the genome of the buffalo parasite.

The automated annotation of the assembly with GeneMark-ES identified new potential genes in regions without reference Muguga gene structures transferred with GMAP to the Buffalo_3081 assembly. In total, 19 genes represented additional copies of *T. parva* genes present elsewhere in the genome, typically in close proximity to their homologs in *T. parva*, based on top BLAST hits. They encoded mostly hypothetical proteins, plus several putative integral membrane proteins. An additional group of six non-syntenic genes were most similar to homologs found in *Theileria annulata* or *Theileria orientalis* (S9 Table), but not found in *T. parva*. Those with best matches to *T. annulata* included hypothetical proteins, a tRNA-pseudouridine synthase I, and a mitochondrial ribosomal protein S14 precursor gene. The single-copy gene with a best match to *T. orientalis* aligned to a region of chromosome 2 of *T. orientalis* (strain Shintoku) with no genes currently annotated.

These 25 new potential genes described above were run through several prediction software packages for further characterization. There were two predicted proteins with five or greater transmembrane helices, a feature usually associated with transmembrane proteins. There were seven genes predicted to contain signal peptide cleavage sites, and no genes were predicted to be GPI-anchored.

A previous study generated whole genome sequence data for a buffalo-derived *T. parva* isolate, referred to as Buffalo LAWR [28]. Interestingly, when we mapped reads from that isolate against the Buffalo_3081 genome assembly, we identified 87,837 SNPs, suggesting that the *T. parva* population associated with buffalo is very diverse genetically, with nearly as many SNPs differentiating buffalo-derived strains as those found in comparisons between genotypes from buffalo *vs*. cattle. This read mapping resulted in an average coverage of 34X of the 25 potential new genes encoded in the genome of *T. parva* Buffalo_3081, similar to the average genome coverage of 27X, supporting their existence.

## Sequence variant detection by read mapping and assembly comparison

To determine if the reconstruction of *de novo* draft genome assemblies with the capture data results in additional power to study rapidly evolving antigens compared to the more common

read mapping-based approach, we identified sequence variants with both methods. Alignment of sequence reads generated for the two non-reference cattle-derived clones, Marikebuni and Uganda, against the reference genome, followed by stringent SNP calling and filtering, yielded 40,228 and 40,835 SNPs. For the buffalo-derived isolate, Buffalo_3081, sequence variant calling resulted in 91,840 SNPs (Fig 2). These values are based on a very high proportion of the genome (S3 Table). These numbers of SNPs are similar to those detected in other cattle-derived isolates in a previous study, which also used a read mapping based approach [28].

A greater number of SNPs was detected by aligning each new assembly to the reference genome and identifying sequence variants (S2 Fig). A total of 55,421 SNPs and 52,385 SNPs were detected in the Marikebuni and Uganda genome assemblies, respectively. Similarly, there were also more SNPs detected in the Buffalo_3081 isolate (n = 124,244) based on assembly comparison, than those detected by read mapping. In each case, ~25% more SNPs were identified by assembly comparison relative to read mapping.

The discrepancy in SNP counts is due to the fact that reads fail to map to their orthologous regions when the sequence is highly variable between genomes, while these loci are nevertheless present in the *de novo* assembly and can be readily compared across strains. This is further supported by our estimates of gene coverage by each of the methods. In each of the non-reference genotypes, the read mapping approach had, on average, worse gene coverage than the respective coverage using assembly generation (Fig 4; S5 Table). Overall, 98.1% of the nuclear genes were recovered in their entirety in BV115 using both read mapping and assembly alignment. In Marikebuni and Uganda the value was 94.7% and 94.8%, respectively, and 93.6% the Buffalo_3081 strain. Among the 56 nuclear genes not fully recovered in all four genotypes,

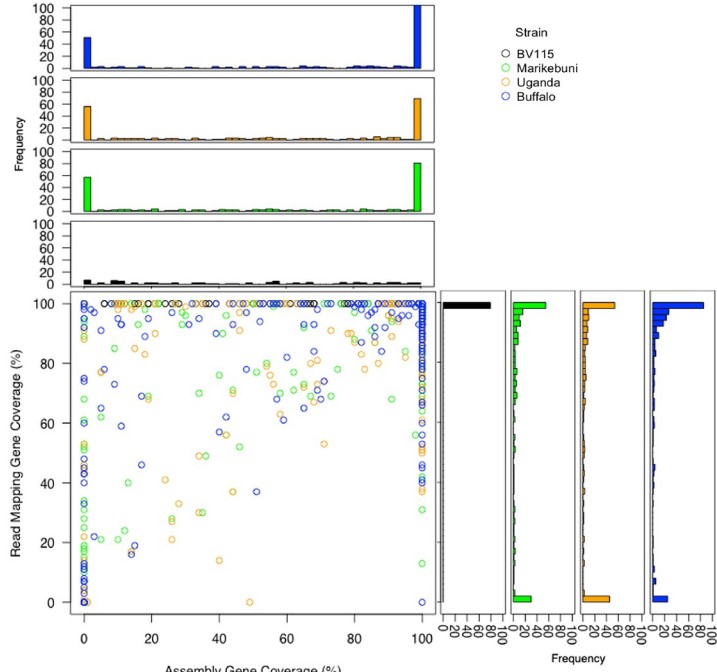

**Fig 4. Read mapping and assembly alignment gene coverage in partially covered genes.** For each gene of the 4,094 nuclear genes not exhibiting 100% coverage when using both read mapping and assembly alignment to the Muguga reference, the percent of the gene covered using each alignment method is shown in the scatter plot for each of the four isolates. The histograms along the top and right side of the plot show the distribution of gene coverage for read mapping and assembly alignment, respectively. The number of nuclear genes covered 100% using both alignment methods were: BV115–4,015; Marikebuni—3,876; Uganda—3,883; Buffalo_3081–3,831.

there were no known single-copy antigens. Most of these genes were located in sub-telomeric regions that are composed of highly repetitive DNA sequences and genes belonging primarily to multigene families (Fig 3), while the rest were members of the *Tpr* gene family that occur in the highly repetitive *Tpr* locus of chromosome 3, and hence more difficult to capture since we did not retain probes that mapped to more than five locations in the genome.

## Variation in protein-coding regions

The high degree of completeness of the genome assemblies makes it possible to obtain estimates of the rates of non-synonymous and synonymous polymorphisms per site, $\pi_N$ and $\pi_S$ respectively, for almost all genes in the genome. This could not be done before due to incomplete read mapping across highly divergent orthologs [35]. We calculated $\pi_N$ and $\pi_S$ among the cattle-transmissible isolates (Uganda, Marikebuni and the reference, Muguga), as well as between the Buffalo_3081 isolate and the reference Muguga (S10 Table). Even though estimates of $\pi_N$ and $\pi_S$ (within species polymorphism) are not reliable estimators of the substitutions rates quantified by $d_N$ and $d_S$ [68], especially for very small values of $\pi$, the relative magnitude of $\pi_N$ across genes and the ratio $\pi_N/\pi_S$, when defined, may still be informative for the identification of rapidly evolving proteins [67, 69, 70].

Among cattle genotypes, the median $\pi_N$ and $\pi_S$ across all protein-coding genes were 0.1% and 1.4%, respectively, and the corresponding values for the divergence between a cattle (Muguga) and a buffalo (Buffalo_3081) strain were 0.6% and 6.1%, respectively (S11 Table). As expected, the ratio $\pi_N/\pi_S$ is lower in the Muguga-Buffalo_3081 comparison, since natural selection has had more time to remove slightly deleterious mutations compared to the comparison among cattle strains (Fig 5A). Despite the elimination of a relatively higher number of deleterious mutations between Muguga-Buffalo_3081 than among cattle strains, $\pi_N$ is still slightly higher in the former comparison because the most recent common ancestor (MRCA) of the Muguga-Buffalo_3081 strains is older than the MRCA of the cattle genotypes (Fig 5B).

Among 37 genes that have been identified as antigens, the average difference in non-synonymous sites between Buffalo_3081 and the ortholog in the reference Muguga was 2.4%, but with a wide range among genes, from 0 to >20%, and a standard deviation of 4.4% (Table 2;

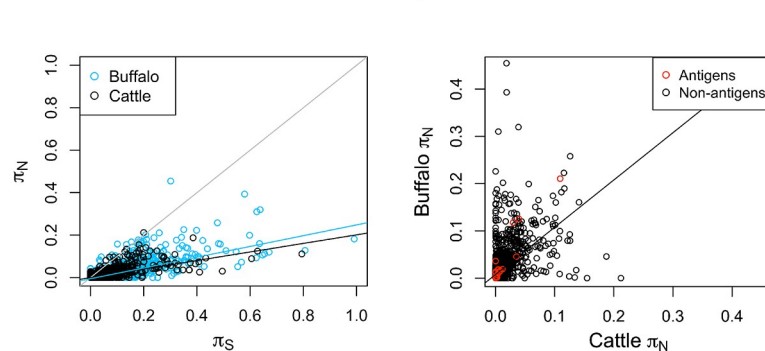

**Fig 5. Nucleotide diversity in amino acid-changing ($\pi_N$) and silent ($\pi_S$) sites. a.** Rates of nucleotide diversity in non-synonymous ($\pi_N$) and synonymous ($\pi_S$) sites for all genes in the Muguga reference genome, both among the cattle strains (Muguga, Marikebuni and Uganda), and between the reference Muguga and the Buffalo_3081. The black regression line corresponds to π value for genes among cattle, the light blue regression line corresponds to the π value for genes among buffalo, and the black line represents y = x. **b.** For each gene, $\pi_N$ among cattle strains (X axis) is compared with $\pi_N$ in a buffalo-derived strain relative to the reference Muguga, (Y axis). Known antigens are colored red. The estimated slope regression line is shown, with a slope of 0.99. The R-squared value for the regression is 0.282. Note the shorter axes in 5b.

**Table 2. Antigen sequence recovery in *T. parva* "Buffalo_3081" and divergence relative to Muguga.**

| Locus tag[1] | Product Name | Functional Annotation | Gene length[2] (bp) | Read mapping coverage[3] (%) | Assembly alignment coverage[4] (%) | Muguga *vs.* Buffalo_3081 $\pi_N$ (%) | Gene rank Muguga-Buffalo_3081 $\pi_N$ (n = 4076) |
|---|---|---|---|---|---|---|---|
| TpMuguga_01g00056 | Tp2 | Hypothetical protein | 1422 | 80.17 | 100 | 12.5 | 41 |
| TpMuguga_01g00188 | Tp6 | Prohibitin | 1104 | 100 | 100 | 0 | 3387 |
| TpMuguga_01g00320 | 11E | Glutaredoxin-like protein | 956 | 100 | 100 | 0.3 | 2769 |
| TpMuguga_01g00726 | Tp16 | translation elongation factor EF-1 subunit alpha | 1316 | 100 | 100 | 1.3 | 1008 |
| TpMuguga_01g00868 | Tp3 | Hypothetical protein | 856 | 100 | 100 | 1.2 | 1120 |
| TpMuguga_01g00939 | gp34 | Schizont surface protein | 1067 | 100 | 100 | 0.7 | 1776 |
| TpMuguga_01g01056 | p32 | 32 kDa surface antigen | 1139 | 100 | 100 | 4.5 | 228 |
| TpMuguga_01g01074 | Tp14 | Haloacid dehalogenase-like hydrolase | 1420 | 100 | 100 | 1.3 | 962 |
| TpMuguga_01g01077 | Tp17 | Haloacid dehalogenase-like hydrolase | 1138 | 100 | 100 | 0.6 | 1976 |
| TpMuguga_01g01078 | Tp21 | Haloacid dehalogenase-like hydrolase | 1189 | 100 | 100 | 0.7 | 1740 |
| TpMuguga_01g01081 | Tp22 | Haloacid dehalogenase-like hydrolase | 1376 | 100 | 100 | 1.3 | 977 |
| TpMuguga_01g01082 | Tp23 | Haloacid dehalogenase-like hydrolase | 1274 | 100 | 100 | 0.4 | 2371 |
| TpMuguga_01g01182 | Tp24 | Lactate/malate dehydrogenase | 1704 | 100 | 100 | 0.1 | 3198 |
| TpMuguga_01g01225 | Tp25 | SVSP family protein | 1671 | 100 | 100 | 2.2 | 541 |
| TpMuguga_02g00010 | Tp26 | SVSP family protein | 1404 | 99.00 | 0 | 0.3 | NA |
| TpMuguga_02g00123 | Tp32 | DEAD/DEAH box helicase | 1692 | 100 | 100 | 0 | 3388 |
| TpMuguga_02g00140 | Tp8 | Cysteine proteinase | 1523 | 100 | 100 | 0.2 | 3010 |
| TpMuguga_02g00148 | X88 | Heat shock protein | 2239 | 100 | 100 | 0 | 3389 |
| TpMuguga_02g00243 | Tp27 | Heat shock protein homolog pss1 | 3048 | 100 | 100 | 0.1 | 3287 |
| TpMuguga_02g00244 | Tp7 | HSP 90 | 3091 | 100 | 100 | 0.1 | 3256 |
| TpMuguga_02g00767 | Tp5 | Translation initiation factor eIF-1A | 939 | 100 | 100 | 0 | 3390 |
| TpMuguga_02g00895 | Tp9 | Hypothetical protein | 1197 | 44.53 | 100 | 21.0 | 6 |
| TpMuguga_02g00958 | Tp28 | SVSP family protein | 2097 | 100 | 100 | 3.6 | 297 |
| TpMuguga_03g00210 | Tp4 | T-complex protein 1 subunit eta | 2763 | 100 | 100 | 0.1 | 3164 |
| TpMuguga_03g00263 | Tp33 | Hypothetical protein | 2605 | 100 | 100 | 1.1 | 1167 |
| TpMuguga_03g00287 | p67 | p67 protein | 2175 | 100 | 100 | 3.6 | 295 |
| TpMuguga_03g00655 | Tp13 | Hypothetical protein | 1476 | 100 | 100 | 0.3 | 2774 |
| TpMuguga_03g00849 | Tp1 | Hypothetical protein | 1822 | 100 | 100 | 1.9 | 633 |
| TpMuguga_03g00861 | p150 | p150 microsphere antigen | 4749 | 100 | 100 | 1.6 | 793 |
| TpMuguga_04g00051 | PIM | Polymorphic immunodominant molecule | 1559 | 65.84 | 100 | 11.8 | 52 |
| TpMuguga_04g00164 | - | Tash protein PEST motif family protein | 1808 | 99.95 | 100 | 7.8 | 116 |
| TpMuguga_04g00437 | p104 | 104 kDa antigen | 3143 | 100 | 100 | 1.8 | 656 |
| TpMuguga_04g00683 | Tp29 | 78 kDa antigen | 2653 | 100 | 100 | 0.1 | 3370 |
| TpMuguga_04g00752 | Tp30 | Ribosomal protein S27a family protein | 664 | 100 | 100 | 0.3 | 2828 |
| TpMuguga_04g00772 | Tp10 | Coronin | 2402 | 100 | 100 | 0.1 | 3305 |

(*Continued*)

**Table 2.** (Continued)

| Locus tag[1] | Product Name | Functional Annotation | Gene length[2] (bp) | Read mapping coverage[3] (%) | Assembly alignment coverage[4] (%) | Muguga *vs.* Buffalo_3081 $\pi_N$ (%) | Gene rank Muguga-Buffalo_3081 $\pi_N$ (n = 4076) |
|---|---|---|---|---|---|---|---|
| TpMuguga_04g00916 | Tp15 | SVSP family protein | 1765 | 100 | 100 | 1.7 | 714 |
| TpMuguga_04g00917 | Tp31 | SVSP family protein | 1647 | 100 | 100 | 2.0 | 590 |

[1]The genes are listed in alphanumerical order, by chromosome and gene order.

[2]Gene length of the reference Muguga allele.

[3]Proportion of Muguga allele with mapping of *T. parva* Buffalo_3081 reads.

[4]Proportion of Muguga allele aligned to the ortholog from the *T. parva* Buffalo_3081.

S12 Table). Three antigens identified previously as highly polymorphic [71] are the ones with the highest $\pi_N$ values between Buffalo_3081 and the reference, namely Tp2, Tp9 and PIM (Table 2). As has been show in other Apicomplexa [72], most polymorphisms are found in genes likely to be involved in host-pathogen interactions, such as those listed above with the highest polymorphisms in Table 2, along with Tp1 and p67. Antigen p67 in particular has been shown to be highly conserved among cattle-derived *T. parva*, but demonstrate polymorphisms in buffalo-derived *T. parva* [73]. Our ability to capture all antigens with near complete coverage using at least one of the approaches demonstrates the value of the capture approach. The ability to fully sequence these antigens in the cattle-and buffalo-derived populations will be useful if future vaccine development against *T. parva* moves toward subunit vaccines [74], as well as understanding how polymorphism levels in specific genes play a role in interactions with drugs that might target them.

To identify rapidly evolving genes, we looked for patterns among the 200 (~5%) genes in three classes: *i*) the highest $\pi_N$ value among cattle genotypes, (*ii*) the highest $\pi_N$ value between cattle (Muguga) and buffalo (Buffalo_3081) genotypes, or (*iii*) the highest $\pi_N/\pi_S$ between cattle (Muguga) and buffalo (Buffalo_3081), the most divergent strains. In all three classes, ~90 genes were annotated as hypothetical, 53 of which were present in two classes and 16 were present in all three classes (S13 Table).

Gene families that were most variable in all three classes included SVSP family proteins, Tpr family proteins, and genes annotated as putative integral membrane family proteins. It is not surprising that the SVSP family genes would have high $\pi$ values, regardless of host, given their localization in sub-telomeric regions and high level of repetitiveness, which are often correlated with higher levels of sequence variation [66]. Likewise, the *Tpr* genes have high levels of sequence variation; this is consistent with their classification as rapidly evolving, antigenic proteins [67, 75]. A number of previously identified antigens also appeared on the list, including Tp1 [76], which demonstrated a high $\pi_N/\pi_S$ ratio, and Tp2 and Tp9, which appeared in all three classes. These results are consistent with recent findings which demonstrated that, of ten previously studied *T. parva* antigens, these three antigens were observed to have the highest level of nucleotide diversity [77].

When comparing only $\pi_N$ of genes among cattle to $\pi_N$ of genes relative to buffalo, the estimated slope was 0.99, showing that the relative rate of non-synonymous divergence across genes is similar in the two populations (Fig 5B). However, an $r^2$ value of 0.282 shows that the regression is not very predictive. Some genes deviate considerably from the regression line, a pattern consistent with differential, host-specific selection in cattle vs. buffalo (Fig 5B), possibly warranting functional investigation. As expected, known antigens are among the most rapidly evolving genes.

## Genomic divergence between cattle- and buffalo-derived *T. parva*

Hayashida and colleagues [28] demonstrated that a greater sequence divergence existed between buffalo-derived genotypes and Muguga (103,880–121,545 SNPs), compared to what was observed among cattle-derived isolates (34,814–51,790 SNPs). The same study found that there was no significant evidence for recombination between cattle- and buffalo-derived *T. parva*, the two populations perhaps having evolved a genetic barrier to recombination. Such a barrier may be due to the absence of piroplasm stages in cattle infected with buffalo-derived genotypes and hence the lack of opportunity for co-transmission of cattle- and buffalo-derived parasites in the same tick [78].

To determine genetic differentiation between cattle- and buffalo-derived *T. parva*, and potentially unusually differentiated loci associated with host adaptation, we estimated Wright's $F_{ST}$, which estimates the amount of genetic variation in the population that is due to differences between the two subgroups. $F_{ST}$ varies between 0 (panmictic population) and 1 (complete differentiation, with the two subgroups fixed for different alleles at all variable genomic sites). We used the data generated here together with data for distinct strains from Hayashida et al. (2013). This analysis showed a mean genome-wide $F_{ST}$ value of 0.436 (S3 Fig). The distribution of $F_{ST}$ varies considerably across each of the four nuclear chromosomes, with some regions in the genome nearly fixed for different variants and others homogeneous among cattle- and buffalo-derived strains (S4 Fig). Approximately 3,000 sites (0.036% of the genome), distributed throughout the genome, were found to have an $F_{ST}$ value that reached genome-wide significance. These patterns of divergence are consistent with an evolutionary old divergence and extensive population differentiation, preventing the identification of specific genes involved in host-specific strain adaptation. Interestingly, though, this extent of divergence in allele frequency does lend support to the assertion that these are two distinct, host-associated parasite populations. An $F_{ST}$ analysis comparing isolates from different studies, but collected from the same host, resulted in an $F_{ST}$ value of 0, ensuring that our comparison between host-associated genotypes is not due to methodological biases of data collected in different studies. These conclusions are somewhat limited by the relatively low genome sequence coverage for samples collected in previous studies (Hayashida *et al.* 2013). A more conclusive study on this topic will require higher coverage from multiple cattle and buffalo *T. parva* genotypes. It is worth noting that previous studies using variable number tandem repeats suggested that *T. parva* populations in co-grazing cattle and buffalo, in central Uganda, were essentially distinct [25].

## Apicoplast genomes

The apicoplast genome from the Muguga reference strain is 39,579 base pairs long and contains 70 genes. The probe set covered 64 of the genes either fully or partially (S1 Table). Aligning sequence reads of all three non-reference isolates achieved complete coverage of all 70 genes encoded by the Muguga apicoplast genome. The assemblies also contained the nearly full gene complement of the Muguga apicoplast, as all 70 genes were covered completely in Marikebuni, and 68 were covered completely in the Uganda and Buffalo_3081 apicoplast assemblies. The two partially covered genes in the Uganda strain apicoplast are TpMuguga_05g00034 (37%) and TpMuguga_05g00040 (93%). In the apicoplast of Buffalo_3081, TpMuguga_05g00037 was covered partially (66%) and TpMuguga_05g00036 was not covered at all.

This fairly complete apicoplast gene set allowed the evaluation of nucleotide diversity per gene. Among cattle isolates, the mean $\pi_N$ and $\pi_S$ values were both 0.2% (S11 Table), less than half the mean values for any of the four nuclear chromosomes. The $\pi_N$ value between the buffalo isolate and the reference was 0.4%, and the $\pi_S$ value was 2.1% (S10 Table). As expected,

these values are greater than the nucleotide diversity among cattle strains, but still considerably lower than the mean nucleotide diversity in nuclear genes among buffalo strains. This suggests that apicoplast genes evolve slower than nuclear genes. Given the role of the apicoplast genes in basic metabolic processes [79], they are expected to be highly conserved. Indeed, high levels of apicoplast sequence conservation have been observed across several Apicomplexa species [80].

## Discussion

This study demonstrates that high quality whole genome sequence data can be generated for intracellular protozoan-infected mammalian cell lines, despite the large excess of host DNA, using a custom oligonucleotide genome capture array. This approach can be costly, adding between $100 and $500 per sample, depending on the kit size (number of reactions purchased), the degree of multiplexing (how many samples captured per reaction), and possible labor costs added to library preparation due to the capture reaction. Additional sequences of *T. parva* have been published since the original *T. parva* reference genome was published in 2005 [29], including those using pyrosequencing technology on DNA extracted from piroplasm-rich blood from infected animals [65], or from lysis of infected lymphocytes [28]. However, these alternatives have limitations. The first approach is not feasible on a large scale, due to animal costs, ethical considerations, and the extensive labor required. The second approach utilizes a schizont purification method from infected cells that has been shown to successfully recover intact parasites, but reports relatively high proportions of host DNA, where at most 80% of reads generated mapped to the *T. parva* reference, increasing sequencing costs and at least partially offsetting the costs associated with capture. In contrast, in the current study, we demonstrate a high level of specificity, where greater than 96% of reads map to the *T. parva* reference, and present an approach that is both high-throughput and less labor-intensive than the alternatives. Future studies should test the applicability of this approach directly to clinical samples (biopsies), obviating the need to culture in lymphocytes, and to tick salivary gland material, which would greatly facilitate the characterization natural variation in the *T. parva* population without the need to passage through cattle.

The whole genome capture method described in the present study, with close to 100% specificity and sensitivity, also represents an advancement in targeted enrichment when compared to approaches previously applied to apicomplexan parasites. An implementation of hybrid selection using whole genome "baits" for the parasite *Plasmodium falciparum* yielded an average of 37-fold enrichment with unamplified samples, and no samples with >50% parasite DNA [81]. A previous application of DNA capture to the apicomplexan *Plasmodium vivax* had a maximum specificity of 80% and sensitivity as low as 84.7% [37]. Three key differences between the present and previous studies are (*i*) the length of the sheared DNA (here >450 bp, but for example, only 200 bp in [37]), (*ii*) the small probe length (here ~76 bp, compared to 140 bp in [81]), and (*iii*) inclusion of probes that map both coding and non-coding regions (e.g., only probes to exons were included in [81]). The first point was intended to facilitate the capture of rapidly evolving or highly variable genomic segments flanking those more conserved and, consequently, matching the probes. The second aimed to maximize the proportion of genome with probe hybridization. Finally, the inclusion of probes to exons as well as introns and intergenic regions was intended to maximize genome recovery and enable genome assembly. Extreme nucleotide composition, such as close to 100% AT content in some intergenic regions and introns in *P. falciparum*, will limit the applicability of these strategies.

The sequences reported here represent the first whole genome datasets from *T. parva* with sufficient quality and depth of coverage to allow the generation of *de novo* genome assemblies

from DNA extracted from infected lymphocyte cultures. This opens a new area for high-throughput genotyping of *T. parva* field isolates of both cattle and buffalo origin, and potentially those isolated from the tick vector. The design of the current probe set eliminated probes whose sequence was found five or more times in the reference genome. As a result, multigene families make up a large portion of the genes not recovered, especially those in the *Tpr* locus, which contains copies with a high degree of sequence similarity[29]. Attempts to improve upon the current results might exclude restrictions of probe copy number representation, and also consider using unique sequences from other *T. parva* strains, including those derived from buffalo. Given that the function of the *Tpr* gene, which comprises one of the largest multigene families in *T. parva*, is unknown [29], the ability to study this gene family is important. While a number of *Tpr* orthologs have been identified in other *Theileria* species [82–84], little is known about this multigene family beyond its 3′ end conserved domain, which contains several transmembrane helices, suggesting that it is an integral membrane protein [75].

As we have shown, *de novo* genome assembly allows the in-depth characterization of genetic polymorphism, which can provide novel insights into population variation and the evolution of this parasite and enable the study of rapidly evolving proteins and protein families of interest. Our study reveals tremendous genetic polymorphism between *T. parva* genotypes, even among just those that are cattle-derived. The average nucleotide diversity among cattle-derived *T. parva* (~6.5 SNPs/kb) is higher than the 4.23 to 6.29 SNP/kb reported before [28]. This is likely attributed to the high sensitivity and specificity of the approach used here, and the resulting ability to reconstruct fairly complete draft genome assemblies. The SNP density observed is also considerably higher than that seen among strains of the malaria-causing apicomplexan *P. falciparum*, of ~1 to 2.3 SNP/kb [85, 86]. This result is explained by the long co-evolution of *T. parva* with the African Cape buffalo, its asymptomatic carrier, dating back millions of years [87]. Therefore, its most recent common ancestor is much older than that of *P. falciparum*, which likely emerged as a result of a relatively recent host transfer from gorilla [88]. The existence of a ECF vaccine that is effective against cattle-transmitted *T. parva*, despite the greater SNP density among cattle-derived *T. parva* strains than observed among *P. falciparum* or even *P. vivax* strains, is highly encouraging, as it suggests that *Plasmodium* genetic diversity *per se* is not an insurmountable obstacle to the development of an effective vaccine.

The generation of the first genome assembly for a buffalo-derived *T. parva* strain presents an improvement upon the previous sequencing of a buffalo strain. Here we generated 250 bp-long reads, which enabled the generation of a draft genome assembly, while before 36 bp read data was generated, and the strain genotyped by read mapping against the reference [28]. This allows us to address, with a high degree of certainty, several long-standing questions in the field. For example, previous studies based both on discrete loci and low coverage genome-wide data [27, 28], suggested that cattle-derived *T. parva* is significantly less diverse than the buffalo-derived *T. parva* population, an assertion supported by our study, which is based on whole-genome sequence data with very high depth of coverage. We also determined that genome size variation exists between cattle- and buffalo-derived parasites, and identified novel genes in the genome of a buffalo-derived strain, which account for a large proportion of its longer overall genome assembly. It is also noteworthy that the majority of the open reading frames unique to Buffalo_3081 were additional copies of genes present in the *T. parva* reference genome. At this point it remains unknown if this is simply due to the whole-genome capture approach used, which is limited to the probes based on the reference genome and their flanking regions, or if in fact *T. parva* (and eukaryotic parasite genomes in general) have a fairly closed pan genome, in which the acquisition of novel genes is very rare and new gene coding sequences (CDSs) are, instead, primarily the outcome of gene duplications and other gene family expansions, followed by rapid sequence divergence [89, 90]. A larger sample size

of genomes analyzed will facilitate the exploration of this issue and elucidate whether the observed pattern is due to gene duplication in buffalo-derived *T. parva* or gene loss in cattle-derived strains.

Characterization of buffalo-derived parasites was necessary to start to identify differences between the two subsets of parasites. Based on this limited sample size, the genome-wide $F_{ST}$ value of 0.436 hints at a strong differentiation between the two subpopulations of *T. parva*. This $F_{ST}$ value is much higher than observed between *P. falciparum* populations sampled across Africa, which ranged between 0.01–0.11 [91]. We also identified a high SNP density between the buffalo isolate and the Muguga reference—nearly double the SNPs detected among cattle isolates, as well as a large number of structural variants. Adding to this genetic evidence of separate host-associated populations are several epidemiological clues [reviewed in 92]. These include the fact that tick transmission of buffalo-derived *T. parva* to other cattle has only been achieved on a few occasions and at low efficiency [93, 94], and previously described immunological observations that differentiate the two parasites [95, 96]. The description of new Apicomplexa species based on genetic information alone is controversial [97, 98] but not unprecedented [99], and it has been proposed that species identification based only on DNA characterization would be more efficient, even though inclusion of other data, such as host and geography, when available, is advisable [100]. Given the mounting evidence of genomic, immunological and epidemiological differences between cattle- and buffalo-derived *T. parva*, we posit that it is appropriate to return to the original classification of these two parasite populations as separate subspecies [13, 101], or perhaps even discuss their classification as separate species. Future studies, based on a representative sample of both cattle and buffalo parasites, will facilitate a comprehensive characterization of the differences between them and the identification of mutations that resulted in novel host-parasite interactions, facilitating adaptation to new mammalian hosts, in the genus *Bos* [102].

Vaccination remains the most cost-effective tool for prevention of livestock infections and concomitant cattle morbidity and mortality. Apicomplexan parasites of livestock are often closely related to human-infective species with respect to the protective immune responses induced, and therefore represent potential models for evaluation of responses to human infection [103]. However, while there are several veterinary vaccines against protozoa that have been manufactured by veterinary authorities in collaboration with the private sector for decades, there is still no fully efficacious vaccine against any protozoan parasites that infect humans [104]. A primary impediment to the development of anti-parasitic vaccines is the high degree of antigenic polymorphism, which results in allele- or genotype-specific efficacy, making the collection of this information critical [105, 106]. Genotype-specificity may also impact the efficacy of vaccines against *T. parva* [107], and resulted in the inclusion of three genotypes in the Muguga Cocktail vaccine [102] which, despite broadly effective against cattle *T. parva*, still does not protect against parasites circulating in buffalo [20]. As a result, it has been proposed that new vaccines should comprise a mixture of antigenically divergent clones [74], the characterization of which our capture approach makes feasible as a high-throughput process. A reference genome for a buffalo-derived *T. parva* parasite will enable a more accurate characterization of genetic variation among buffalo-derived strains, in particular in genomic regions that are unique to buffalo parasites or highly divergent from cattle-derived strains. The characterization of multiple buffalo-derived genotypes will reveal the proportion of the genome that is variable among these strains and differences that are fixed relative to cattle-derived genotypes, These data can be used to inform a decision on whether a single vaccine that protects both against cattle and buffalo *T. parva* sub-populations is feasible. This is particularly pertinent given the high level of genetic differentiation we report between buffalo-derived *T. parva* and the reference, cattle-derived, Muguga genome.

The problem is particularly acute when the pathogen parasitizes nucleated host cells, resulting in an extremely small ratio of host-to-parasite DNA [36]. This presents a substantial obstacle to whole genome characterization for species in the genus *Theileria*, as well as for bacterial pathogens such as *Chlamydia* and *Rickettsia* [36, 108, 109]. The approach described here offers a major advance in the capacity to characterize genetic diversity of intracellular protozoan parasite populations, which potentially can enhance informed development of more broadly efficacious vaccines, including protective vaccines against buffalo-derived *T. parva*. To achieve the widest protection, any future development of subunit vaccines against *T. parva* should consider the inclusion of orthologs from buffalo-derived strains.

In conclusion, the potential applications of the capture approach on *T. parva* samples are many and will be valuable in answering translational questions, including improving vaccine design and understanding breakthrough infection by buffalo-derived genotypes in vaccinated cattle [20], as well as understanding the impact of parasite genetic variation on the efficacy of potential drugs. Characterization of bovine-infecting genotypes can also be used to understand the role of heterologous reactivity, or infection with multiple *Theileria* species, that has been implicated as a determinant of the impact of disease control measures at the population level [110]. Finally, the availability of multiple genome sequences may shed light on the mechanism and frequency of host switching from buffalo to cattle that led to the establishment of the two distinct parasite populations described in this study.

## Supporting information

**S1 Fig. Generation of the field derived *T. parva* stabilates used in this study.** Stabilate numbers are shown. An * indicates stabilates generated using a 3-step cloning procedure as described by Morzaria and colleagues [64]. Stabilates used in this study are colored in yellow. The last step of the preparation of stabilates (except for 3121.11) included tick pick-up on infected cow or buffalo [64].
(PNG)

**S2 Fig. Total count of single nucleotide polymorphisms (SNPs) per sample.** The total number of SNPs identified was compared between read-mapping and assembly approaches, for each of the four isolates. As expected, almost no SNPs were found between BV115 (animal infected with the Muguga strain) and the reference *T. parva* Muguga genome assembly. Identification of SNPs based on assembly alignment is consistently more sensitive than red mapping. Twice as many SNPs are found in the buffalo- than in cattle- derived strain relative to the Muguga reference, from cattle.
(PNG)

**S3 Fig. Genetic differentiation between cattle- and buffalo-derived strains.** Window-based $F_{ST}$ analysis comparing cattle strains to those derived from buffalo. Analysis includes our strains and those from Hayashida *et al.* (2013). The windows used were 4,000 bp long with a 1,000 bp overlapping window. Genome-wide $F_{ST}$ was 0.436.
(PNG)

**S4 Fig. Distribution of $F_{ST}$ values within nuclear chromosomes.** Histograms of window-based $F_{ST}$ values calculated for each nuclear chromosome, showing a wide range of $F_{ST}$ values throughout the genome. Frequency of average $F_{ST}$ value is shown on x-axis, $F_{ST}$ values shown on y-axis.
(PNG)

**S1 Table. Probe Coverage Statistics.** Properties of gaps in probe coverage across all assembly contigs and supercontigs (chromosomes) and across protein-coding genes.
(DOCX)

**S2 Table. Sample metadata.** Properties related to library construction and generation of whole genome sequence data.
(DOCX)

**S3 Table. Sequence variants identified by read mapping relative to the reference *T. parva* Muguga genome assembly.**
(DOCX)

**S4 Table. Assembly validation and correction.**
(DOCX)

**S5 Table. Gene content in *de novo T. parva* genome assemblies.** Assembly length, number of contigs and gene content are shown. Gene content was assessed with two approaches: "read mapping" and "assembly alignment".
(DOCX)

**S6 Table. Homology searches for predicted *Theileria parva* genes in unmapped contigs.**
(DOCX)

**S7 Table. Structural variants in *de novo* assemblies compared to reference assemblies.**
(DOCX)

**S8 Table. Structural variants between *T. parva* strains.**
(DOCX)

**S9 Table. Best BLAST match for each *T. parva* Buffalo_3081 gene without a detectable homolog in *T. parva* Muguga.**
(DOCX)

**S10 Table. Nucleotide diversity among cattle-derived *T. parva* strains (Muguga, Marikebuni and Uganda), and between cattle (Muguga reference) and the buffalo-derived strain *T. parva* Buffalo_3081.**
(XLSX)

**S11 Table. Summary statistics of nucleotide diversity among cattle-derived *T. parva* strains, and between cattle and the buffalo-derived strain *T. parva* Buffalo_3081 shown in Table S10.**
(DOCX)

**S12 Table. Summary statistics of nucleotide diversity among cattle-derived *T. parva* strains, and between cattle and the buffalo-derived strain *T. parva* Buffalo_3081 for known *T. parva* antigens.**
(DOCX)

**S13 Table. Detection of rapidly evolving genes.**
(DOCX)

## Acknowledgments

All sequencing was done at the Institute for Genome Sciences' Genomics Resource Center.

## Author Contributions

**Conceptualization:** Nicholas C. Palmateer, W. Ivan Morrison, Claudia A. Daubenberger, Vish Nene, Richard P. Bishop, Joana C. Silva.

Data curation: Olukemi O. Ifeonu, Hanzel T. Gotia, Joana C. Silva.

Formal analysis: Nicholas C. Palmateer, Kyle Tretina, Olukemi O. Ifeonu.

Funding acquisition: Vish Nene, Donald P. Knowles, Joana C. Silva.

Investigation: Nicholas C. Palmateer, W. Ivan Morrison, Joana C. Silva.

Methodology: Nicholas C. Palmateer, Joshua Orvis, Luke Tallon, Joana C. Silva.

Project administration: James B. Munro.

Resources: Roger Pelle, Elias Awino.

Software: Joshua Orvis, Jonathan Crabtree, Elliott Drabék.

Supervision: Joana C. Silva.

Validation: Joshua Orvis.

Visualization: Nicholas C. Palmateer, Jonathan Crabtree, Elliott Drabék.

Writing – original draft: Nicholas C. Palmateer, Richard P. Bishop, Joana C. Silva.

Writing – review & editing: Nicholas C. Palmateer, Kyle Tretina, Jonathan Crabtree, W. Ivan Morrison, Claudia A. Daubenberger, Donald P. Knowles, Richard P. Bishop, Joana C. Silva.

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
