## [Decision Letter · Decision Letter 0]

27 Jul 2020

Dear Dr. Silva,

Thank you very much for submitting your manuscript "Capture-based enrichment of Theileria parva DNA enables full genome assembly of first buffalo-derived strain and reveals exceptional intra-specific genetic diversity" for consideration at PLOS Neglected Tropical Diseases. As with all papers reviewed by the journal, your manuscript was reviewed by members of the editorial board and by several independent reviewers. The reviewers appreciated the attention to an important topic. Based on the reviews, we are likely to accept this manuscript for publication, providing that you modify the manuscript according to the review recommendations. 

Please ensure that deposition of sequence data to accessible repositories is complete, as explained in the submission guidelines.

Sincerely,

Andrew Paul Jackson, Ph.D.

Guest Editor

Mary Lopez-Perez

Deputy Editor

Editor's comments:

line 165; can you not tell from the SNPs in the Muguga_3087 and Buffalo_7014 genome sequences whether they are clonal or not?

line 208; please explain why it is necessary to limit sequence coverage, by sub-sampling reads 

line 184; The nimblegen oligo library was designed against the reference, what is the sequence identity tolerance of this with respect to successful hybridisation? 

line 274-9: High sensitivity relates to how strains sequence map to the reference, but what about strain sequence that is missing from the reference? You quantify sequence that assembles de novo without mapping to the reference, but what about sequences that were not captured at all? Are you able to compare assemblies of 454 and illumina reads for the same DNA?

line 700; please comment on what could be done to adapt the oligo array design to ensure tpr and other variable regions are captured from clinical samples

Table2; I think the table should be ordered somehow to help the reader, perhaps on decreasing variability

line 578; "Morrison in prep", if reference not in print then please provide a citable DOI or arXiv URL.

line 608; "Palmateer in prep", if reference not in print then please provide a citable DOI or arXiv URL.

Reviewer's Responses to Questions

**Key Review Criteria Required for Acceptance?**

**Methods**

-Are the objectives of the study clearly articulated with a clear testable hypothesis stated?

-Is the study design appropriate to address the stated objectives?

-Is the population clearly described and appropriate for the hypothesis being tested?

-Is the sample size sufficient to ensure adequate power to address the hypothesis being tested?

-Were correct statistical analysis used to support conclusions?

-Are there concerns about ethical or regulatory requirements being met?

Reviewer #1: The objectives and study design was clearly presented. I have no issues.

Reviewer #2: (No Response)

Reviewer #3: The methodology appears appropriate - see comments

**Results**

-Does the analysis presented match the analysis plan?

-Are the results clearly and completely presented?

-Are the figures (Tables, Images) of sufficient quality for clarity?

Reviewer #1: Results and figures are clear and well presented. 

"Raw sequence reads and corresponding genome assemblies available from NCBI under project number

779 PRJNA16138. (In progress)" - please ensure that all assemblies are indeed available. I could only see the T. Parva Muguga strain genome - maybe I missed it though.

Reviewer #2: (No Response)

Reviewer #3: see comments

**Conclusions**

-Are the conclusions supported by the data presented?

-Are the limitations of analysis clearly described?

-Do the authors discuss how these data can be helpful to advance our understanding of the topic under study?

-Is public health relevance addressed?

Reviewer #1: It seems that a disadvantage to the DNA-capture approach is the failure to cover members of multigene families – please discuss this a bit further. The probe set used for DNA capture covers >95% of the genome, and only 53 genes were completely without probe coverage. Of those 53 genes, many belong to multigene families. I would like to see a bit more discussion about the significance of the Tpr (and SVSP and other) multigene families in Theileria biology – in particular what is known about the biological function of Tpr multigene families, and to what extent have Tpr loci been sequenced in other Theileria isolates / species / following different methods. 

The authors allude to a the high level of genetic divergence between cattle and buffalo derived T. parva having “clear implications for vaccine development” – I’d appreciate a bit more detail here and a more in-depth discussion of what these implications are.

Reviewer #2: (No Response)

Reviewer #3: see comments

**Editorial and Data Presentation Modifications?**

Reviewer #1: Line 260: “The capture probe set targets almost completely span the nuclear and apicoplast genomes…» - delete either “targets” or “Span”

Reviewer #2: (No Response)

Reviewer #3: see comments

**Summary and General Comments**

Reviewer #1: In this manuscript, the authors describe a capture-based target enrichment approach that enabled de novo assembly of almost-complete Theileria parva genomes derived from infected host cell lines. Importantly, the first buffalo-derived T. parva genome assembly is presented and compared with the cattle-derived parasites. A remarkably high level of genetic divergence (as measured by non-synonymous nucleotide diversity per gene) was observed between buffalo and cattle-derived parasites. The capture method uses custom-designed biotinylated oligonucleotides for hybridization to the target sequence – in this case the probes were designed on the T. parva Muguga reference genome. The sensitivity and specificity of this capture method was very impressive. Sequencing was performed using Illumina technology. 

In my opinion the analysis of the new genome assemblies was robust, and the data well laid out and clearly presented. In particular the authors present the first annotated genome assembly from buffalo-derived T. parva – this will be a hugely valuable resource, in particular in the context of vaccine design. I only have a few very minor points.

Reviewer #2: In their study “Capture-based enrichment of Theileria parva DNA enables full genome assembly of first buffalo-derived strain and reveals exceptional intra-specific genetic diversity” the authors describe a DNA capture technique that lead to the assembly and analysis of the buffalo-derived T. parva genome. I have no major issues, but I do have a number of more minor comments that should be considered by authors and editor before final acceptance of the manuscript.

P2L49: How much larger is the genome? Has it also more predicted genes? This information could be added to the abstract. 

P2L52: FST – abbreviation should be explained.

P2L55: What is the advantage and what are the previous methods used (see also comment below)?

P2L59/60 and P3L86/87: I doubt that this number is correct. It appears very high. Please give a timely reference (e.g. in the introduction).

P5L115-125: The procedure using aerolysin (cp. Baumgartner et al. Microbes and Infection 1999) appears to be highly efficient and produces good quality parasite DNA for NGS approaches. This alternative approach should be more critically discussed and compared by the authors. In particular as it was successfully used for NGS approaches before (cp. Hayashida et al. 2013).

P6L164: Why was the buffalo and Muguga strain not cloned prior to analysis? Most field isolates are multiclonal. Given that high SNP ratios can be observed in particular in highly variable genes such as SVSPs between strains, are multiclonal isolates not complicating SNP analyses?

 P6L165/166: Since Hayashida et al. (2013 DNA Research) have already sequenced and analysed the same buffalo-derived strain, it appears important to further stress the advantage of the current study (such as the genome assembly).

P14, Figure 2: Legends and annotations are in part too small.

Reviewer #3: see comments

PLOS authors have the option to publish the peer review history of their article (what does this mean?). If published, this will include your full peer review and any attached files.

Reviewer #1: No

Reviewer #2: No

Reviewer #3: No
---

## [Editor Report · Decision Letter 1]

8 Sep 2020

Dear Dr. Silva,

We are pleased to inform you that your manuscript 'Capture-based enrichment of Theileria parva DNA enables full genome assembly of first buffalo-derived strain and reveals exceptional intra-specific genetic diversity' has been provisionally accepted for publication in PLOS Neglected Tropical Diseases.

Best regards,

Andrew Paul Jackson, Ph.D.

Guest Editor

Mary Lopez-Perez

Deputy Editor

---

## [Editor Report · Acceptance letter]

20 Oct 2020

Dear Dr. Silva,

We are delighted to inform you that your manuscript, "Capture-based enrichment of *Theileria parva* DNA enables full genome assembly of first buffalo-derived strain and reveals exceptional intra-specific genetic diversity," has been formally accepted for publication in PLOS Neglected Tropical Diseases.

Best regards,

Shaden Kamhawi

co-Editor-in-Chief

Paul Brindley

co-Editor-in-Chief
